# Antioxidants Amelioration Is Insufficient to Prevent Acrylamide and Alpha-Solanine Synergistic Toxicity in BEAS-2B Cells

**DOI:** 10.3390/ijms241511956

**Published:** 2023-07-26

**Authors:** Hoda Awad Eltayeb, Leandra Stewart, Mounira Morgem, Tommie Johnson, Michael Nguyen, Kadeshia Earl, Ayodotun Sodipe, Desirée Jackson, Shodimu-Emmanuel Olufemi

**Affiliations:** 1Department of Biology, Texas Southern University, Houston, TX 77004, USA; 2Department of Environmental and Interdisciplinary Sciences, Texas Southern University, Houston, TX 77004, USA

**Keywords:** butylated hydroxyanisole (BHA), butylated hydroxytoluene (BHT), acrylamide, alpha (α)-solanine, BEAS-2B cells, AKT/PKB

## Abstract

Cells produce free radicals and antioxidants when exposed to toxic compounds during cellular metabolism. However, free radicals are deleterious to lipids, proteins, and nucleic acids. Antioxidants neutralize and eliminate free radicals from cells, preventing cell damage. Therefore, the study aims to determine whether the antioxidants butylated hydroxyanisole (BHA) and butylated hydroxytoluene (BHT) will ameliorate the maximum dose of acrylamide and alpha (α)-solanine synergistic toxic effects in exposed BEAS-2B cells. These toxic compounds are consumed worldwide by eating potato products. BEAS-2B cells were simultaneously treated with BHA 10 μM and BHT 20 μM and incubated in a 5% CO_2_ humidified incubator for 24 h, followed by individual or combined treatment with acrylamide (3.5 mM) and α-solanine (44 mM) for 48 h, including the controls. Cell morphology, DNA, RNA, and protein were analyzed. The antioxidants did not prevent acrylamide and α-solanine synergistic effects in exposed BEAS-2B cells. However, cell morphology was altered; polymerase chain reaction (PCR) showed reduced RNA constituents but not DNA. In addition, the toxic compounds synergistically inhibited AKT/PKB expression and its downstream genes. The study showed BHA and BHT are not protective against the synergetic toxic effects of acrylamide and α-solanine in exposed BEAS-2B cells.

## 1. Introduction

Cells of living organisms are constantly exposed, directly and indirectly, to environments that are said to be toxic or non-toxic. Exposure of living organisms to toxic chemical compounds in these environments is deleterious to cell survival because these chemical compounds can derail many of the cellular processes [1,2,3]. These harmful chemical compounds induce free radicals, which cause damage to parts of the cell, such as the cell membrane, nucleic acid, and protein, by stealing their electrons through oxidation. Antioxidants enrich cell survival by scavenging and decomposing free radicals. However, the mechanisms by which antioxidants improve the cellular response to free radicals in the presence of toxicants/toxins are not well understood. Antioxidants ameliorate the effects of free radicals in cells through multiple mechanisms that reduce the occurrence of different human disorders provoked by free radicals [4,5,6]. Also, it has been reported that acrylamide causes intracellular overproduction of free radicals [7], and it increases the production of more free radicals by depleting the production of endogenous antioxidants in the colonic tissue [8]. Solanine was shown to increase the production of ROS, which caused the cell death of human hepatoma G2 (HepG2) cells [9]. Therefore, it is significant to investigate how antioxidants ameliorate the combined effects of two potato byproducts: acrylamide (2-Propenamide), a toxicant, and alpha-solanine (α-solanine), a toxin. They are toxic to cells [10,11], but their synergistic effects in cells must be better understood and studied in the presence or absence of combined antioxidants. 

By the Maillard reaction, consumable foods such as ripe black olives [12], barley grains [13], baked cookies [14], potato chips [15], and roasted coffee [16] cooked at higher temperatures formed acrylamide. Acrylamide possesses cytotoxic and genotoxic properties. Acrylamide is genotoxic and induces oxidative stress that causes oxidative DNA damage in HepG2 cells [17], as well as the failure of other partners of the antioxidant defense system [18]. In addition, acrylamide induces morphological transformation in Syrian hamster embryo (SHE) cells [10]. The molecular mechanism(s) of how acrylamide morphologically transforms cells are not well understood. The events leading to acrylamide-induced carcinogenesis are also not well understood to date, with several inconsistencies in the published data, most pointing at the reactive metabolite glycidamide [19]. The acrylamide neurotoxicity effect is well established in laboratory animals [20,21]. Acrylamide is transformed into glycidamide in cells during metabolic increases in DNA damage in mice and rat spermatocytes [22]. Glycidamide forms an adduct with the DNA, and it is the source of DNA damage [23]. In addition, acrylamide increases reactive oxygen species (ROS) that cause DNA breakage [17,24,25].

Alpha-solanine (α-solanine) and alpha-chaconine (α-chaconine) are natural toxins. They are glycoalkaloids found in potatoes, but only α-solanine is often studied. They are species of the Solanaceae family (e.g., potatoes, tomatoes, eggplants, and peppers), including the non-Solanaceae family (e.g., tobacco, petunia, and climbing or bittersweet nightshade). They are stressed metabolites, or phytoalexins, that protect the Solanaceae and the non-Solanaceae families from predators and pathogens. In potatoes, glycoalkaloid content increases after pathogenic attacks, which increases phytoalexin (i.e., antimicrobial) function [26]. Pathogenic attacks also promote the synthesis of antioxidants in the affected area of the potatoes [27]. Glycoalkaloids possess both toxic and anti-carcinogenic properties, and they elicit their actions via the disruption of cholesterol-containing cell membranes that cause leakage of the cell content [27]. In HepG2 cells, α-solanine lowers mitochondrial membrane potential via mitochondrial membrane permeability transition (MPT), which increases Ca^2+^ released by mitochondria into the cytoplasm to promote apoptosis [28]. Accumulation or a high dose of glycoalkaloids is required to be harmful in humans [29], and a high amount causes death in hamsters [30]. Glycoalkaloids and antioxidants are found in potatoes, which possess anti-oncogenic properties. In addition to glycoalkaloids found in potatoes, potato peels contain antioxidant and antimicrobial properties attributed to their phenolic compounds [31] and phytoalexin properties [32,33]. In normal and cancer cells, glycoalkaloids behave similarly (as anti-oncogenes and toxicants) and could inhibit cell proliferation and promote cell death [34].

Antioxidants (BHA; 2(3)-t-butyl-4-hydroxyanisole) and butylated hydroxytoluene (BHT; 2,6-di-tert-butyl-4-methyl phenol) are synthetic phenolic compounds. BHA and BHT are added to foods to prevent rancidity and preserve them. BHA prevents hydrogen peroxide (H_2_O_2_) apoptosis in primary cultured mouse hepatocytes by impeding ROS production [35], and BHT, as well as other antioxidants, inhibits H_2_O_2_ in *Saccharomyces cerevisiae* [36]. Even though antioxidants may possess possible tumor-promoting and anti-carcinogenic properties, at a low dose, BHA and BHT are not carcinogenic [37]. BHA and BHT are phenolic antioxidants with chemoprevention properties that inhibit free radicals from inducing normal cells into cancerous cells [38]. Oxidative stress promotes an imbalance between oxidants and antioxidants to disrupt redox signaling and control, leading to molecular damage in cells due to the production of ROS/ RNS in cells.

The PI3/AKT pathway regulates genes involved in cell survival and apoptosis, and it is one of the most frequently dysregulated pathways in cancer [39]. Oxidative stress causes DNA damage, base modifications, abasic sites, and strand breaks [40,41]. On the other hand, RNA is more vulnerable to oxidative stress damage [42]. The effects of oxidative stress on mitochondrial DNA (mtDNA) may be less severe since mitochondria generate low ROS and have an efficient antioxidant defense system [43].

AKT is a serine-threonine protein kinase, also known as protein kinase B (PKB), a crucial signaling molecule that controls the balance between cell survival and apoptosis. The role of AKT in DNA damage is well studied [44]. AKT promotes unperturbed cell cycles by acting on diverse downstream factors controlling the transitions of G1/S and G2/M. Other PI3K-like kinases (PIKKs), ATM, ATR, and DNA-PK, can activate AKT [45], which modulates DNA damage and genome stability responses through several downstream factors.

This study focused on the toxic effects of acrylamide and α-solanine on BEAS-2B cells in the presence of antioxidants. In this study, BEAS-2B cells were initially treated with antioxidants, BHA and BHT, simultaneously and incubated in a 5% CO_2_ humidified incubator for 24 h. Then, the cells were exposed to acrylamide alone, α-solanine alone, or acrylamide/α-solanine and incubated continuously for 48 h. Also included in this study are control cells. This study showed that the antioxidants did not alleviate the cell morphological change and altered RNA and protein expressions induced by acrylamide and α-solanine on BEAS-2B cells. In addition, DNA alteration was not observed with the PCR method employed. Reduced RNAs of miRNA stem-loop (i.e., hsa-Let-7c stem-loop) and protein phosphatase 2A (PP2A) were observed, suggesting that the chemicals caused reduced RNA in BEAS-2B cells, and they should also be tested in different types of cells to determine if the RNA reduction is uniform in other cells.

## 2. Results

### 2.1. Cell Morphology and Protein Expression of AKT and Its Downstream Genes

This study shows that pretreatment of BEAS-2B cells with BHA/BHT did not prevent acrylamide and α-solanine from altering the morphology of the cells compared to the controls. Simultaneous treatment of the chemicals affected the morphology of the cells more than their individual treatment or half dose.

#### 2.1.1. Effects of Antioxidants, Toxicants, and Toxins on Cell Morphology

Cell morphological changes were not observed in untreated BEAS-2B cells: untreated control at 24 h (Figure 1A), untreated control at 48 h (Figure 1B), and treated control combined with BHA (10 µM)/BHT (20 µM) for 24 h/PBS (1X) for 48 h (Figure 1G). Furthermore, BHA, BHT, and PBS do not affect the morphology of the cells because the cells (Figure 1A,B,G) display normal epithelial-like growth and proliferation. However, the number of untreated control cells at 24 h (Figure 1A) was less than the untreated control at 48 h (Figure 1B) and treated-control cells at 48 h (Figure 1G). The increase in the number of cells (Figure 1G) is due to the 48 h incubation time. There were dead cells in the untreated controls (Figure 1A,B) and treated control (Figure 1G) cells, but they were less noticeable.

Figure 1 consists of BEAS-2B cells analyzed under different conditions for morphological changes. The cells were labeled A, B, C, D, E, F, and G. Untreated control cells at 24 h (Figure 1A), treated control cells at 48 h (Figure 1B), and cells treated with BHA (10 µM)/BHT (20 µM) for 24 h plus PBS (1X) for 48 h (Figure 1G) have normal cell morphology and proliferation. However, BEAS-2B cells treated with BHA (10 µM)/BHT (20 µM) for 24 h plus acrylamide (3.5 mM) for 48 h have morphological changes. They have irregular cell body shapes and elongated structures with long pseudopodia that connect the cells. Morphological changes observed among these cells were crenation (green arrow), shrunken (red arrow), and apoptosis (orange arrow) cells (Figure 1C).

Cells treated with BHA (10 µM)/BHT (20 µM) for 24 h plus α-solanine (44 mM) for 48 h (Figure 1D) have cells that form mesh-like structures. The cell body is irregular and has elongated structures with long pseudopodia connecting each cell to other cells. Similar morphological changes were observed, as in Figure 1C.

Figure 1E shows cells treated with BHA (10 μM)/BHT (20 μM) for 24 h plus combined acrylamide (3.5 mM) and α-solanine (44 mM) for 48 h were affected more than other treated cells because the cell amounts reduced due to cell death, which is the synergistic effect of acrylamide and α-solanine on the BEAS-2B cell morphology. Morphological changes observed among these cells were crenation (green arrow), shrunken (red arrow), and apoptosis (orange arrow). Cell loss due to cell death proves that the combined effect of acrylamide and α-solanine is harmful to the BEAS-2B cells.

At half dose, cells treated with BHA (10 μM)/BHT (20 μM) for 24 h plus combined acrylamide ½(3.5 mM) and α-solanine ½(44 mM) 48 h had morphological changes (Figure 1F) that were not as severe as the total dose of acrylamide and α-solanine at 48 h. The cells also formed colonies with irregular cell bodies. They have elongated structures with long pseudopodia connecting cells in each colony to others.

#### 2.1.2. Change in Nucleic AcidsChanges in DNA

##### Change in RNA

Figure 2 shows a contrast image of amplified genomic DNA (gDNA) PCR bands separated on a 2.5% TAE agarose electrophoresed gel stained with ethidium bromide. The three primer sets, D2S123 (Figure 2A), AKT2 (Figure 2B), and MT-CO1 (Figure 2C: mitochondrial DNA (mtDNA)), were used for PCR amplification. The amplified PCR bands were used to determine if there were differences in the amplified DNA bands between experimental samples and controls. Differences in the amplified PCR bands were measured based on the pixel intensity values, but no difference existed for the three primer sets. The sequences of the DNA primer sets are:D2S123 (Forward Primer: 5′-AAACAGGATGCCTGCCTTTA-3′ and Reverse Primer: 5′-GGACTTTCCACCTATGGGAC-3′ (>chr2: 51,061,299 + 51,061,509, size 211 bp).AKT2 (Forward Primer: 5′-CTTTGTCATACGCTGCCTGCAGT-3′ and Reverse Primer: 5′-TCTCCTCACACCAGGCTTGCTC-3′ (>chr19: 40,255,100 + 40,255,229, size 130 bp).Mitochondrial CO1 (MT-CO1) (Forward Primer: 5′-GGAGCTTTGGCAACTGACT-3′ and Reverse Primer: 5′-CTGCTAGGTGTAAGGAGAAGATGG-3′ (>chrM: 6130 + 6363, size 234 bp).

Figure 2A–C amplified DNA bands did not reveal any changes in the DNA. The contrast gel electrophoresis gel shows amplified DNA of the experimental samples (3–7) and controls (1, 2, 8, and 9), including a marker (M) and a one-kilobase (1 kb) ladder. The amplified DNA bands produced by the D2S123, AKT2, and MT-CO1 primers are 211 bp, 130 bp, and 234 bp, respectively. The PCR results of the amplified DNA bands for the untreated and treated samples showed similar intensity, including control gDNA, suggesting no changes in the DNA contents of the cells or no DNA damage in the primer regions.

##### Change in RNA

The hsa-Let-7c expression was determined based on the amplified PCR product. Figure 3 shows a contrast gel image with bands of amplified hsa-Let-7c stem-loop PCR products analyzed on a 2.5% TAE agarose electrophoresis gel stained with ethidium bromide (Figure 3A) and visualized in the LI-COR Odyssey Fc Imaging System. The bands were quantified in the LI-COR Image Studio Lite Software Version 5.2 to generate the trim signal values of the amplified bands displayed as a bar graph (Figure 4B, 1–8)—the trim signal values were used to quantify the amount of expressed hsa-Let-7c. The amplified hsa-Let-7c PCR bands are organized and ordered on the agarose gel (Figure 3A, 1–9) and the bar graph (Figure 3B, 1–9). The experimental samples are 3 through 6, and the controls are 1, 2, 7, 8, and 9.

A bar graph shows the quantification values of the amplified hsa-Let-7c stem-loop (Figure 3B), including the controls, gDNA, and water (H_2_O). The hsa-Let-7c PCR samples are organized and ordered in the agarose gel and bar graph (Figure 3B, 1–9). The hsa-Let-7c primer set consists of a forward and a reverse primer:The hsa-Let-7c primer set (Forward Primer: 5′-GTTGTATGGTTTAGAGTTACAC-3′ and Reverse Primer: 5′-GCTCCAAGGAAAGCTAGAAGGTT-3′ (>chr21: 16,539,849 + 16,539,911, size 63 bp).

Figure 3B shows a bar graph corresponding to the hsa-Let-7c bands (Figure 3A), which denote the amount of expressed hsa-Let-7c. The bar graphs are the measurements of expressed hsa-Let-7c in the experimental samples (Figure 3B, 3–6) and controls (Figure 3B, 1, 2, 7), including genomic DNA (gDNA) and H_2_O controls (Figure 3B, 8, 9). Expression of hsa-Let-7c gradually decreased in the experimental samples (Figure 3B, 3–6) compared to controls (Figure 3B, 1, 2) and was further reduced in the treated PBS control (Figure 3B, 7) compared to the experimental samples (Figure 3B, 3–5). Samples of cells pretreated with antioxidants, BHA (10 μM)/BHT (20 μM) for 24 h and simultaneously treated with acrylamide ½(3.5 mM) and α-solanine ½(44 mM) for 48 h showed a drastically reduced hsa-Let-7c (Figure 3B, 6) compared to other treated experimental samples (Figure 3B, 3, 4, 5) and other controls (Figure 3B, 1, 2, 7, 8). The result suggests that the pretreatment of the cells with antioxidants, BHA (10 μM)/BHT (20 μM) for 24 h, is not protective against the toxicant or toxin when present individually or synergistically.

PP2A RNA expression was determined using amplified PP2A cDNA bands synthesized from total RNA. Figure 4A shows a contrast agarose gel image with bands of amplified PP2A PCR products analyzed on a 2.5% TAE agarose electrophoresis gel stained with ethidium bromide. Figure 4B displays a corresponding bar graph of a semi-quantification of amplified PP2A cDNA PCR products, including a water (H_2_O) control (Figure 4A, 8 and Figure 4B, 8) used for validation and contamination detection. The PP2A primer set consists of a forward and a reverse primer:The PP2A primer set (Forward Primer: 5′-CAAATGGAAGCGTTCTCAGGCATAC-3′ and Reverse Primer: 5′-TTCCTCATGAACCTCATTCCACATCTC-3′ (cDNA NM_178001.3 and chr9: 129,111,415 − 129,148,946) (>chr9: 129,111,711 + 129,111,841, size 130 bp).

The trim signal values measured as the amounts of PP2A expression are the pixel intensity values trimmed by 5%, highest and lowest, and were produced in the LI-COR Image Studio Lite Software Version 5.2. The trim signal values of the amplified cDNA products are displayed as a bar graph (Figure 4B, 1–7, and 8 (control water with no template)) used to quantify the amount of PP2A RNA expression. PP2A expression decreased (Figure 4A, 3–6 and Figure 4B, 3–6) compared to controls (Figure 4A, 1, 2 and Figure 4B, 1, 2), including PBS (Figure 4A, 7 and Figure 4B, 7). Pretreatment of BEAS-2B cells with BHA (10 μM)/BHT (20 μM) for 24 h plus acrylamide (3.5 mM) and α-solanine (44 mM) for 48 h did not prevent reduced PP2A (Figure 4A, 5 and Figure 4B, 5), suggesting the synergistic toxic effect of acrylamide and α-solanine altered PP2A RNA expression. At half the dose, acrylamide ½(3.5 mM) and α-solanine ½(44 mM), the synergistic toxic effect is minimal compared to higher amounts of acrylamide (3.5 mM) and α-solanine (44 mM) (Figure 4A, 5 and Figure 4B, 5). Furthermore, acrylamide (3.5 mM) alone reduced PP2A RNA expression (Figure 4A, 3 and Figure 4B, 3) compared to α-solanine (44 mM) alone (Figure 4A, 4 and Figure 4B, 4), suggesting acrylamide alone reduces PP2A RNA expression more than α-solanine. In addition, PBS (1X) (Figure 4A, 7 and Figure 4B, 7) did not reduce PP2A RNA expression compared to controls at 24 h (Figure 4A, 1 and Figure 4B, 1) but slightly higher than control at 48 h (Figure 4A, 2 and Figure 4B, 2), suggesting PP2A RNA is less vulnerable than smaller mRNA (e.g., miRNAs) because PP2A RNA is large and capable of forming a secondary RNA structure, which may likely reduce its vulnerability. 

#### 2.1.3. Protein Expression of AKT and Its Downstream Genes

Figure 5A shows the expression of ACTB, AKT, BcL-xL, Bax, CASP3, and CASP9 proteins on chemiluminescent western blots detected by the LI-COR Odyssey Biosciences Fc Chemiluminescence Imaging System. LI-COR Image Studio Lite Software Version 5.2 determined the pixel intensity values of the protein bands. The trim signal values of the proteins are displayed as a bar graph (Figure 5B, 1–7) with error bars and quantified trim signal values. The control is Beta (β)-actin (ACTB). The experimental samples are numbered 3 through 6, and the control samples are numbered 1, 2, and 7 (Figure 5A,B).

Figure 5A shows the results of the Western blots, and ACTB and AKT bands were evident in all samples (Figure 5A, 1–7). Still, the intensity of their protein bands varies within and among the samples. The band intensities of ACTB and AKT decreased in the sample pretreated with BHA (10 μM)/BHT (20 μM) for 24 h and simultaneously treated with acrylamide (3.5 mM) and α-solanine (44 mM) for 48 h (Figure 5A, 5), suggesting that the chemicals affected ACTB and AKT expressions (Figure 5A, 5). In addition, to address the variability of ACTB and AKT protein expression, it was apparent that the untreated control sample at 48 h and the sample pretreated with BHA (10 μM)/BHT (20 μM) for 24 h plus PBS (1X) have bands with higher intensity within their group (Figure 5A, 2 and 7) compared to the sample pretreated with BHA (10 μM)/BHT (20 μM) for 24 h plus a sample treated with toxic chemicals for 48 h (Figure 5A, 3) or plus a sample treated with toxic chemicals for 48 h (5A, 6). These treated cells expressed an average amount of protein compared to the control at 48 h (Figure 5A, 2), and the sample pretreated with BHA (10 μM)/BHT (20 μM) for 24 h plus PBS (1X) treated for 48 h (Figure 5A, 7). The results suggest that the chemicals affect ACTB, a housekeeping protein, and AKT, a cell survival regulatory protein. Figure 5A, 1, control at 24 h, has less than average ACTB and AKT expression, suggesting that reduced expression of ACTB and AKT in the toxic chemicals treated cells was due to chemical toxicity on the cells.

Bcl-xL expression occurred in all samples but varied (Figure 5A, 1–7). Bcl-xL band intensities were more pronounced in control at 48 h (Figure 5A, 2), sample pretreated with BHA (10 μM)/BHT (20 μM) for 24 h plus α-solanine (44 mM) treated for 48 h (Figure 5A, 4), and sample pretreated with BHA (10 μM)/BHT (20 μM) for 24 h plus PBS (1X) (control) treated for 48 h (Figure 5A, 7), compared to control at 24 h (Figure 5A, 1), sample pretreated with BHA (10 μM)/BHT (20 μM) for 24 h plus acrylamide (3.5 mM) treated for 48 h (Figure 5A, 3), and sample pretreated with BHA (10 μM)/BHT (20 μM) for 24 h plus combined acrylamide (3.5 mM) and α-solanine (44 mM) treated for 48 h (Figure 5A, 5). These samples have low band intensities. Also, samples pretreated with BHA (10 μM)/BHT (20 μM) for 24 h plus acrylamide ½(3.5 mM)/α-solanine ½(44 mM) treated for 48 h (Figure 5A, 6) expressed mild Bcl-xL compared to samples (Figure 5A, 2, 4, 7) that expressed a higher level of Bcl-xL protein. The lower band intensity of Bcl-xL in control at 24 h (Figure 5A, 1) shows that the protein is less expressed at 24 h since the cells were not treated compared to treated cells (Figure 5A, 3, 5) that expressed deficient Bcl-xL protein. Also, the differences in Bcl-xL band intensities in Figure 5A, 3, 4, 5, and 6 prove that the chemicals affected Bcl-xL expression variably.

Bax band intensities were not observed in samples pretreated with BHA (10 μM)/BHT (20 μM) for 24 h plus acrylamide (3.5 mM) treated for 48 h (Figure 5A, 3) and samples pretreated with BHA (10 μM)/BHT (20 μM) for 24 h plus combined acrylamide (3.5 mM) and α-solanine (44 mM) treated for 48 h (Figure 5A, 5). The results suggested that acrylamide alone and combined acrylamide and α-solanine affect Bax protein expression. Bax band intensities were weak in samples pretreated with BHA (10 μM)/BHT (20 μM) for 24 h plus α-solanine (44 mM) treated for 48 h (Figure 5A, 4) and samples pretreated with BHA (10 μM)/BHT (20 μM) for 24 h plus acrylamide ½(3.5 mM)/α-solanine ½(44 mM) treated for 48 h (Figure 5A, 6). Bax expression was almost undetectable in the control sample at 24 h (Figure 5A, 1), suggesting the presence of some likely Bax residues at 24 h. Overall, the toxic chemicals affected the expression of the Bax protein. Also, CASP3 and CASP9 proteins were undetected when the cells were treated with antioxidants plus a full dose of acrylamide alone or a combined full dose of acrylamide and α-solanine (Figure 5A, 3 and 5). Even at a combined half dose of acrylamide and α-solanine, the cells did not express CASP3 and CASP9 (Figure 5A, 6), suggesting the chemical affected the expression of these proteins.

Figure 5B shows a bar graph with error bars and a table of trim signal values derived from quantifying the protein bands (Figure 5A) using LI-COR Image Studio Lite Software Version 5.2 to generate pixel intensity values trimmed 5% below and above, producing trim signal values. Each bar corresponds to the amounts of proteins expressed in control and experimental cells. The error bars are the measurement variability, indicating uncertainty or error in the measured values. All the groups had variability in the expressed proteins under different conditions (Figure 5A), supported by the Western blot results (Figure 5B).

Expression of the different proteins in the untreated control cells at 24 h in group 1 was deficient, or none, compared to untreated control cells at 48 h in group 2 and treated control cells at 48 h in group 7 (pretreated cells with BHA (10 μM)/BHT (20 μM) for 24 h, plus PBS (1X) treated for 48 h). Expression of the different proteins in treated cells at 48 h in group 3 (pretreated cells with BHA (10 μM)/BHT (20 μM) for 24 h, plus acrylamide (3.5 mM) treated for 48 h) was deficient or none compared to untreated control cells at 48 h group 2 and untreated control cells at 48 h group 7 (pretreated cells with BHA (10 μM)/BHT (20 μM) for 24 h, plus PBS (1X) treated for 48 h), suggesting that the acrylamide affect the expression of the different proteins variably.

The expression of the different proteins in groups 3, compared to groups 1, 2, and 7, shows that acrylamide affects protein expression variably. Expression of the different proteins in treated cells at 48 h in group 4 (pretreated cells with BHA (10 μM)/BHT (20 μM) for 24 h, plus α-solanine (44 mM) treated for 48 h) was minimal compared to group 2 and untreated control cells at 48 h in group 7 (pretreated cells with BHA (10 μM)/BHT (20 μM) for 24 h, plus PBS (1X) treated for 48 h), suggesting that α-solanine affects the expression of the protein minimally and it is less potent at a dose of 44 mM (Group 4) compared to acrylamide at a dose of 3.5 mM (Group 3).

Expression of the different proteins in treated cells at 48 h in group 5 (pretreated cells with BHA (10 μM)/BHT (20 μM) for 24 h, plus combined acrylamide (3.5 mM) and α-solanine (44 mM) treated for 48 h) was close to none compared to groups 2 and 7, confirming that synergistically acrylamide (3.5 mM) and α-solanine (44 mM) affect the protein expression in the treated cells detrimentally. On the other hand, expression of the different proteins in treated cells at 48 h in group 6 (pretreated cells with BHA (10 μM)/BHT (20 μM) for 24 h, plus combined acrylamide ½(3.5 mM) and α-solanine ½(44 mM) treated for 48 h) was minimal in a few proteins or none in a couple of proteins compared to groups 2 and 7. Still, compared to group 5, it differs, suggesting that the combined treatment of acrylamide (3.5 mM) and α-solanine (44 mM) is more potent than their half doses.

#### 2.1.4. Signal Value of Protein Expression and Statistical Significance Test of One-Way ANOVA: AKT, Bcl-xL, Bax, CASP3, and CASP9

The error bars (Figure 5B) measured the expressed proteins’ variability and indicated uncertainty or error in the measured values. Even though several of the error bars overlapped, signaling evidence of insignificance among the means of the data sets. Yet, we proceeded to confirm if it was true and employed a one-way analysis of variance (ANOVA) to determine if differences in protein expression were significant. For the ANOVA, the ratios of trim signal values of the proteins (i.e., AKT, Bcl-xL, Bax, CASP3, and CASP9) were determined using ACTB as the control. The ratios of trim signal values were employed for ANOVA (Table 1). For the one-way ANOVA, when alpha equals 0.05 (α = 0.05), the one-way ANOVA was insignificant among the group means of the proteins tested: F(4,30) = 1.4731, *p* = 0.2351 (*p* > 0.05). The F critical value is 2.6896, which is higher than the F-statistic value (Table 2).

Since the *p*-value and the F-statistic are insignificant, it suggests that there is no difference between the group means of the proteins and that they cluster together more tightly than the within-group variability. Occasionally, one-way ANOVA may not be significant; it does not mean that there is no difference among the means. They may not be detectable with the mean values of the groups.

## 3. Discussion

Previous studies demonstrated the relevant effects of acrylamide toxicity in in vitro cell culture studies using BEAS-2B cells [46] and other cell lines [47,48,49]. Different IC_50_ concentrations have been used in different cell lines, and the most commonly used IC_50_ value is 5 mM [50]. Therefore, we used a 3.5 mM IC_50_ value of acrylamide in this study since it did not harm the BEAS-2B cells. Also, α-solanine toxicity has been studied in different cell lines at different IC_50_ values~10 μM, 20 μM, 30 μM and 50 μM [34,51,52]. Our experiments discovered that a 44 mM IC_50_ value of α-solanine does not harm the BEAS-2B cells. The toxicity of acrylamide and α-solanine is not studied synergistically, but acrylamide and α-solanine are typically studied separately or combined with or without other chemicals and antioxidants [53,54]. However, the amelioration of acrylamide toxicity by antioxidants is more studied than α-solanine, whose studies focused on its toxicity to kill cancer cells. Still, α-solanine is a steroidal glycoalkaloid found in potatoes, which contain antioxidants that prevent oxidation processes [55], which is likely to make α-solanine harmful. In addition, glycoalkaloids can cause inflammatory bowel disease (IBD) in the mammalian intestine by destroying the intestinal epithelial barrier if excessively fried potatoes or potato skin are consumed [56,57]. Glycoalkaloids have anticarcinogenic and antiproliferative properties that inhibit the proliferation of cancer and normal cells [32,58,59,60,61,62]. Unfortunately, this function of glycoalkaloids is detrimental to normal cells and organisms that ingest foods containing higher levels of glycoalkaloids. We simultaneously used BHA (10 μM) and BHT (20 μM) to ameliorate the synergistic effects of acrylamide and α-solanine in vitro in BEAS-2B cells. The study deciphers the combined impacts of acrylamide and α-solanine in the presence of antioxidants treated in vitro cultured cells; it shows the effects of two chemicals in potatoes, the most commonly consumed food, are detrimental when combined in an in vitro study.

Effects of acrylamide on the morphology of BEAS-2B cells [46] and other cell lines [63,64] have been reported. In addition, α-solanine caused morphological changes in HepG2 cells, resulting in numerous cytoplasmic vacuoles, and the cells becoming round, shrunken, and detached [9]. In this study (Figure 1D), α-solanine in the presence of antioxidants caused BEAS-2B cells to acquire irregular morphological structures such as apoptotic cells, shrunken cells, and crenation cells with non-pronounced pseudopodia structures that connect some cells. Acrylamide affects the morphology of BEAS-2B cells (Figure 1C) more than α-solanine (Figure 1D). Still, the combination of acrylamide and α-solanine in the presence of BHA/BHT (Figure 1E) affects the morphology and loss of BEAS-2B cells (Figure 1E). This study reports the effect of combined acrylamide and α-solanine on BEAS-2B cells in the presence of antioxidants. We are unsurprised by the synergistic effect of potato products, acrylamide, and α-solanine. Still, we are more concerned about the health issues that may develop when potatoes with a higher level of glycoalkaloids and foods rich in acrylamide are consumed excessively.

Several reports show that acrylamide causes mutations in DNA through acrylamide adduct formation [65], and the same is true for glycidamide [65,66,67,68], an acrylamide intermediary breakdown product. On the other hand, solanine has anticancer properties, and its anticancer activity is effective across different tumors [69]. One of the anticancer properties of solanine is that it targets different proteins. Acrylamide’s cytotoxic and genotoxic effects result in apoptosis. Therefore, treating cells with acrylamide and α-solanine plus antioxidants will likely result in apoptosis without genotoxicity-induced DNA damage or acrylamide-induced DNA damage occurring in selected tissues and sites within the DNA [65]. In this study, when primers for D2S123 (i.e., instability detection), *AKT2* (i.e., gene damage detection), and *MT-CO1* (mitochondrial DNA (mtDNA: damage detection)) were used for PCR amplification of genomic DNA, there were no detected differences between amplified PCR products from the control cell and treated cell DNA (Figure 2A–C). The explanations for the lack of mutation detection in our experimental group DNA samples are: (1) the primers are not designed to detect transitions (A:T → T:A) and transversions (G:C → C:G) base changes and single base deletion mutations [70,71]; (2) A deletion or a base pair may not have occurred in the regions of the primers; (3) acrylamide causes DNA adduct in specific tissues or selected DNA regions within the genome of organisms to promote mutations [70,71]. One result that stands out was the treatment of BEAS-2B cells with acrylamide and α-solanine individually or combined in the presence of antioxidants, which altered RNA expression levels (i.e., hsa-Let-7c RNA: Figure 3A, 3–5 and Figure 3B, 3–5 and PP2A RNA: Figure 4A, 3–5 and Figure 4B, 3–5). Still, their profound toxic effects are pronounced when combined (Figure 3A,B and Figure 4A,B). These results suggest that acrylamide and α-solanine affect RNA expression more than DNA damage since RNA is a single molecule and DNA is a double-stranded molecule.

Several pieces of evidence showed that acrylamide and α-solanine altered protein expression levels differentially by targeting similar or dissimilar proteins in pathways that unleashed their individual toxic effects. For example, based on a proteomic neuron toxicity study, acrylamide adducts with protein cysteine (Cys) groups [70]. There is no evidence of α-solanine adduct formation with DNA or protein. However, α-solanine’s antitumor properties strongly inhibit cancer cell growth through RNA expression (i.e., miR-138) [71,72,73], (i.e., miR-18a) [74], and protein [70]. Results of this study showed that acrylamide and α-solanine affect RNA or protein expression of genes in the AKT/PKB pathways (Figure 5). Still, individual effects differ from their synergistic toxic effects at the maximal dose, which is deleterious to BEAS-2B cells (Figure 1E compared to Figure 1F).

Acrylamide and α-solanine, individually or combined in the presence of antioxidants, altered the level of RNA expression (i.e., hsa-Let-7c RNA: Figure 3A,B and PP2A RNA: Figure 4A,B). Acrylamide alone reduced expression of ACTB, AKT/PKB, and Bcl-xL (Figure 5A, 3) compared to untreated control (Figure 5A, 2) and treated control (Figure 5A, 7), and synergistically, acrylamide and α-solanine reduced expression of ACTB, AKT/PKB, and Bcl-xL (Figure 5A, 5) compared to treated control (Figure 5A, 2) and treated control (Figure 5A, 7) but the expression of Bax, CASP3, and CASP9 were not detected (Figure 5A, 3 and 5). In addition, at a combined half dose of acrylamide and α-solanine, the expression of CASP3 and CASP9 were not detected (Figure 5A, 6). Reduced protein expression observed in the experimentally treated cells is likely due to the higher cell death caused by acrylamide alone and the combination of acrylamide and α-solanine exerting their effects on the BEAS-2B cells. On the other hand, α-solanine alone minimally reduced expression of ACTB, AKT/PKB, and Bcl-xL (Figure 5A, 4) compared to untreated control (Figure 5A, 2) and treated control (Figure 5A, 7), suggesting that α-solanine alone affects BEAS-2B minimally compared to acrylamide alone. Also, the combined half dose of acrylamide and α-solanine reduced protein expression of ACTB, AKT/PKB, and Bcl-xL near to minimal (Figure 5A, 6) compared to α-solanine alone (Figure 5A, 4), suggesting that the combined full dose of acrylamide and α-solanine affect the BEAS-2B cells (Figure 5A, 3). The quantification of protein expression was determined by the pixel intensity values of the protein bands counted and expressed as trim signal values, which trimmed 5% of the highest and lowest pixel intensity values. One-way analysis of variance (ANOVA) was used to determine if there are significant differences between the expressed protein pixel intensity values that were converted into trim signal values in the Image Studio Lite Software Version 5.2 of the Biosciences Li-COR Odyssey Fc chemiluminescence imaging system and were used for ANOVA when alpha equals 0.05 (α = 0.05). The F-statistic (1.4731) and *p*-value (0.2351) were insignificant (Table 2).

## 4. Materials and Methods

### 4.1. Cell Culture of BEAS-2B Cells and Cell Exposure

BEAS-2B cells purchased from the American Type Culture Collection (ATCC^®^ CRL-9609, Manassas, VA, USA) are normal human bronchial epithelial cells obtained from a non-cancerous individual’s autopsy. BEAS-2B cells were seeded in a 75 mL culture flask and incubated in a 5% CO_2_ humidified incubator (NuAire Laboratory Equipment, Plymouth, MN, USA) according to the standard cell culture protocol. After the BEAS-2B cells reached 80% confluence, the cells were trypsinized using Gibco™ Trypsin-EDTA (0.05%), and cells were counted via the Bio-Rad TC20 Automated Cell Counter (Bio-Rad, Hercules, CA, USA) and seeded at two million cells per culture dish plate in twenty-one plates of 60 mm × 15 mm. The individual culture dish contained two million cells. BEAS-2B cells in the twenty-one tissue-culture dish plates were divided into three groups. Each group consists of seven tissue-culture dish plates. For cells in the three groups—each dish plate consists of BEAS-2B cells—two of the cell culture dish plates in each group were used as controls and were incubated for 24 h and further incubated in a 5% CO_2_ humidified incubator for 48 h. Five of the cell culture dish plates in each group were initially treated simultaneously with Butylated Hydroxyanisole (BHA: 10 μM) and Butylated Hydroxytoluene (BHT: 20 μM)) (MilliporeSigma, Burlington, MA, USA) and incubated in a 5% CO_2_ humidified incubator for 24 h. Furthermore, they were treated as follows and incubated in a 5% CO_2_ humidified incubator for 48 h: plate 1: Acrylamide (3.5 mM) (MilliporeSigma, Burlington, MA, USA); plate 2: α-solanine (44 mM) (MilliporeSigma, Burlington, MA, USA); plate 3: Combined acrylamide (3.5 mM) and α-solanine (44 mM); plate 4: Combine acrylamide ½(3.5 mM) and α-solanine ½(44 mM); and plate 5: phosphate buffer saline (PBS) (1X) (Thermofisher™, Waltham, MA, USA). The cells were used for DNA, RNA, and protein extractions as follows: 1. DNA was extracted from group one cells; 2. RNA was extracted from group two cells; and 3. Proteins were extracted from group three cells.

### 4.2. Cell Morphological Analysis

After incubation, the control cells and the treated cells were analyzed under the Eclipse Ti-E Inverted Nikon Microscope (Nikon Instruments Inc., Melville, NY, USA) to determine if the cells have undergone morphological changes, such as apoptosis resulting from necrosis, blebbing, crenation, or oncosis.

### 4.3. DNA, RNA, and Proteins Extractions

#### 4.3.1. DNA

The Genomic DNA (gDNA) Purification Kit (Thermofisher™, Waltham, MA, USA) was used to extract DNA from group one BEAS-2B cells. The DNA was extracted according to the manufacturer’s protocol. The optical density of 260/280 (OD_260/280_) was performed for DNA purity. Optical density (OD_260_) was used to determine the amount of each extracted total DNA. Spectronic BioMate 3 UV-Vis (Thermofisher™, Waltham, MA, USA) confirmed the OD_260_ of the individual DNA sample. A polymerase chain reaction (PCR) was employed to determine a change in the DNA content or DNA damage disrupting the phosphodiester bond between adjacent deoxyribose residues in the DNA.

#### 4.3.2. Total RNA

The Total RNA Purification Kit (Thermofisher™, Waltham, MA, USA) was used to extract total RNA from the group two BEAS-2B cells using the TRIzol™ reagent (Thermofisher™, Waltham, MA, USA), according to the manufacturer’s instructions. Cells in the individual tissue-culture dish were treated with 1 mL of TRIzol™ reagent (Thermofisher™, Waltham, MA, USA) to lyse the cells; the lysed cells were transferred into new individual 1.5 mL microcentrifuge tubes containing the cellular materials and gently mixed by vortexing. Chloroform (200 mL) at room temperature was added to the individual lysed TRIzol-cell mixture. Each TRIzol-cell mixture in the 1.5 mL microcentrifuge tube was incubated at room temperature for 5 min. After incubation of the TRIzol-cell mixture at room temperature for 5 min, each TRIzol-cell mixture was centrifuged at 14,000 rpm in a 4 °C microcentrifuge (Precision, Winchester, VA, USA) for 20 min to separate the chloroform and cell debris from the aqueous layer. The aqueous solution layer in the TRIzol-cell mixture was removed and transferred into new 1.5 mL microcentrifuge tubes. Ice-cold isopropanol (500 mL) (MilliporeSigma, Burlington, MA, USA) was added to each aqueous solution layer and incubated at room temperature for 10 min to precipitate the RNA. After incubation at room temperature for 10 min, each aqueous solution-isopropanol mixture was centrifuged at 14,000 rpm in a 4 °C microcentrifuge (Precision, Winchester, VA, USA) for 20 min to collect RNA pellets that bound to the bottom of the microcentrifuge tubes. After the RNA pellet stuck to the bottom of the microcentrifuge tubes, the aqueous isopropanol was decanted, separating the pellet from the aqueous solution. The pellets of the RNA sample were rinsed with 75% ethanol twice and centrifuged twice, and the RNA sample in the individual 1.5 mL microcentrifuge tube was air-dried at room temperature. Each RNA sample was dissolved in Nuclease-Free Water (Thermofisher™, Waltham, MA, USA). The integrity of the individual extracted RNA was determined based on the presence of the ribosomal RNA bands—28S, 18S, and 5S separated on a 1% TAE agarose electrophoresis gel stained with ethidium bromide (EtBr) and visualized the inside Li-COR Odyssey^®^ Fc imager (LI-COR Biosciences, Lincoln, NE, USA) using Image Studio Lite Version 5.2 (LI-COR Biosciences, Lincoln, NE, USA). Optical density (OD_260_) was used to determine the amount of the extracted total RNA. Spectronic BioMate 3 UV-Vis (Thermofisher™, Waltham, MA, USA) confirmed the OD_260_ of the individual total RNA sample.

#### 4.3.3. Protein

For protein extraction of group three BEAS-2B cells, the cells were lysed using mammalian protein extraction reagent, Radio-Immunoprecipitation Assay (RIPA: 1X) lysis buffer supplemented with a phenylmethanesulfonylfluoride (PMSF), protease/phosphate inhibitor (1:100), and protease (1:100) (Cell Signaling, Beverly, MA, USA), according to the manufacturer’s instructions. Cells in the individual tissue-culture dish containing the 1X RIPA lysis buffer were scraped off the plate, transferred into a separate 1.5 mL microcentrifuge tube containing the cellular materials, and gently mixed by tapping multiple times. The lysed cells in the 1.5 mL microcentrifuge tubes were placed on ice for 45 min. After incubation of the cells in ice for 45 min, the cells were centrifuged in a 4 degrees Celsius (4 °C) microcentrifuge (Precision, Winchester, VA, USA) at 14,000 revolutions per minute (rpm) for 20 min to separate the cell debris at the bottom of the tube from the aqueous proteins. The proteins were removed from the individual 1.5 mL microcentrifuge tubes and transferred into new 1.5 mL microcentrifuge tubes. The extracted proteins were stored at −80 °C.

#### 4.3.4. Complementary DNA (cDNA) Synthesis

According to the manufacturer’s protocol, individual extracted total RNA was used to synthesize complementary DNA (cDNA) using the SuperScript™ III First-Strand Synthesis System (Thermofisher™, Waltham, MA, USA). For cDNA synthesis, 5 micrograms (µg) of the individual extracted total RNA was placed in an individual 0.2 mL microcentrifuge tube, and the total RNA was treated with DNAse I and DNAse I 10X Buffer and incubated at 37 °C for 1 h in a C1000 Touch™ Thermal Cycler (Bio-Rad, Hercules, CA, USA). After the DNase I treatment for 1 h, the DNase reaction of the individual total RNA sample was inactivated according to the manufacturer’s protocol. Both random hexamer and oligo (dT) primers in the SuperScript™ III First-Strand Synthesis System (Thermofisher™, Waltham, MA, USA) were used according to the manufacturer’s protocol to synthesize the cDNA inside the C1000 Touch™ Thermal Cycler (Bio-Rad, Hercules, CA, USA). 

#### 4.3.5. Polymerase Chain Reaction (PCR) and Gel Electrophoresis

Polymerase chain reaction (PCR) was used to determine genomic DNA and RNA damage that destroys DNA and RNA phosphodiester bonds and causes DNA or RNA breakage. D2S123 microsatellite and *AKT2* gene primers were used to determine if phosphodiester bond breakage occurred in nuclear DNA. A mitochondrial cytochrome c oxidase subunit I (*MT-CO1*) gene primer set was used to determine if DNA damage occurred in mtDNA, including reduced DNA contents. Human hsa-Let-7c RNA (a microRNA) and PP2A primer sets were used to determine if RNA damage and altered expression occurred in RNA and if the toxicant and/or the toxin affected RNA expression.

For the PCR Reaction Mix: A total volume of the PCR reaction mixture was 15.0 µL (contains 2.5 µL of genomic DNA (5 ng) or cDNA (~25 ng), 7.5 µL of HotStar Taq Master Mix (Qiagen, Valencia, CA, USA), 1.0 µL (100 ng) of forward primer, 1.0 µL (100 ng) of reverse primer (Millipore Sigma, Burlington, MA, USA), and 3 µL of nuclease-free H_2_O (Thermo Fisher Scientific, Waltham, MA, USA)). The PCR reaction mixtures were placed and run inside a C1000 Touch™ Thermal Cycler (Bio-Rad, Hercules, CA, USA). The PCR conditions are as follows: 95 °C for 15 min, 94 °C for 45 s, 60 °C or 55 °C for 30 s, 72 °C for 60 s, 45 cycles, and 72 °C for 10 min.

For gel electrophoresis and analysis, 5.0 µL of the amplified PCR products of the individual experimental sample was mixed with DNA loading dye and electrophoresed on a 2.5% agarose gel stained with ethidium bromide. Each stained agarose gel was placed inside the Li-COR Odyssey^®^ Fc imager (LI-COR Biosciences, Lincoln, NE, USA) to visualize the amplified DNA bands and capture the agarose gel image. The amplified DNA bands were visually analyzed or quantified using a trimmed intensity signal in Image Studio Lite Version 5.2 (LI-COR Biosciences, Lincoln, NE, USA). The trimmed intensity signal in Image Studio Lite Version 5.2 excludes five percent of pixels with the highest and lowest pixel intensities from the total pixel intensity signals.

#### 4.3.6. Statistical Analysis

Statistical significance of a one-way test was employed to determine the mean of the expressed proteins with alpha set at 0.05 (α = 0.05), and the *p*-value (*p* < 0.05) and F-statistic were calculated.

## 5. Conclusions

Antioxidants BHA and BHT did not prevent the toxic effects of acrylamide and α-solanine in BEAS-2B cells. The chemicals affect cell morphology, RNA expression of hsa-Let-7c and PP2A, and protein expression variably at different experimental conditions. We did not detect the effect of acrylamide and α-solanine on DNA based on our detection method. Acrylamide alone is more harmful than α-solanine in the BEAS-2B cells. Simultaneous treatment of BEAS-2B cells with acrylamide (3.5 mM) and α-solanine (44 mM) caused more cell death than their half dose.

Acrylamide (3.5 mM) and α-solanine (44 mM) affect the expression of ACTB, AKT, and Bcl-xL. The chemicals affect ACTB, a housekeeping gene, and one of the genes used in normalizing gene expression of experimental values. The chemicals affect protein expression in different experimental conditions, apart from the controls. Acrylamide alone reduced the expression of proteins that are downstream of AKT. It reduced Bcl-xL expression to near low levels but prevented the expression of Bax, CASP3, and CASP9 compared to untreated and treated controls, while α-solanine alone did not. Acrylamide ½(3.5 mM) and α-solanine ½(44 mM) half doses affect CASP3 and CASP9 more than other proteins. This study was designed to understand the toxic effect of combined acrylamide and α-solanine on in vitro cells.

## Figures and Tables

**Figure 1 ijms-24-11956-f001:**
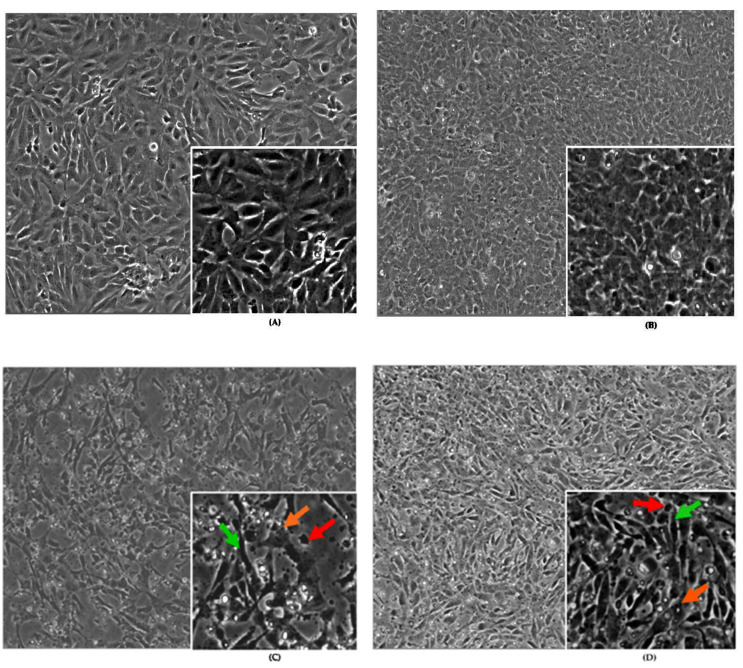
Microscopic observation of cells at 100× (10 × 10) magnification with cropped sections of the cell images: (**A**) Untreated control, 24 h. (**B**) Untreated control, 48 h. (**C**) Treated BHA (10 μM)/BHT (20 μM) 24 h plus acrylamide (3.5 mM) 48 h. (**D**) Treated BHA (10 μM)/BHT (20 μM) 24 h plus α-solanine (44 mM) 48 h. (**E**) Treated BHA (10 μM)/BHT (20 μM) 24 h plus acrylamide (3.5 mM) and α-solanine (44 mM) 48 h. (**F**) Treated BHA (10 μM)/BHT (20 μM) 24 h plus combined acrylamide ½(3.5 mM)) and α-solanine ½(44 mM) 48 h. (**G**) Treated BHA (10 μM)/BHT (20 μM) 24 h plus PBS (1X) 48 h.

**Figure 2 ijms-24-11956-f002:**
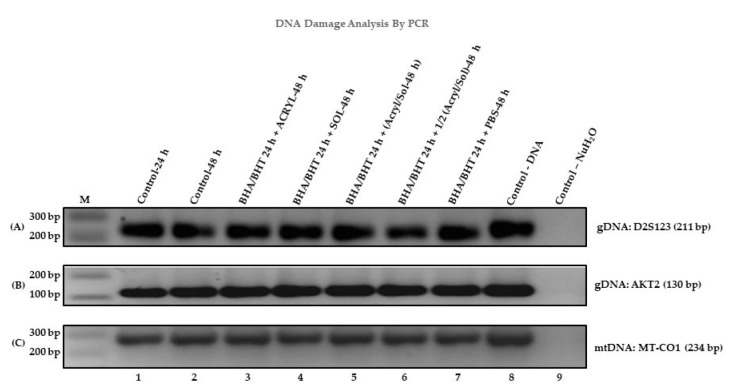
Changes in DNA contents determined by PCR: A contrast image of agarose electrophoresis gel—(**A**) Amplified PCR bands of D2S123. (**B**) Amplified PCR bands of AKT. (**C**) Amplified PCR bands of MT-CO1. The wells were organized in the agarose gel as M (1 kb ladder), and the samples were numbered 1–9.

**Figure 3 ijms-24-11956-f003:**
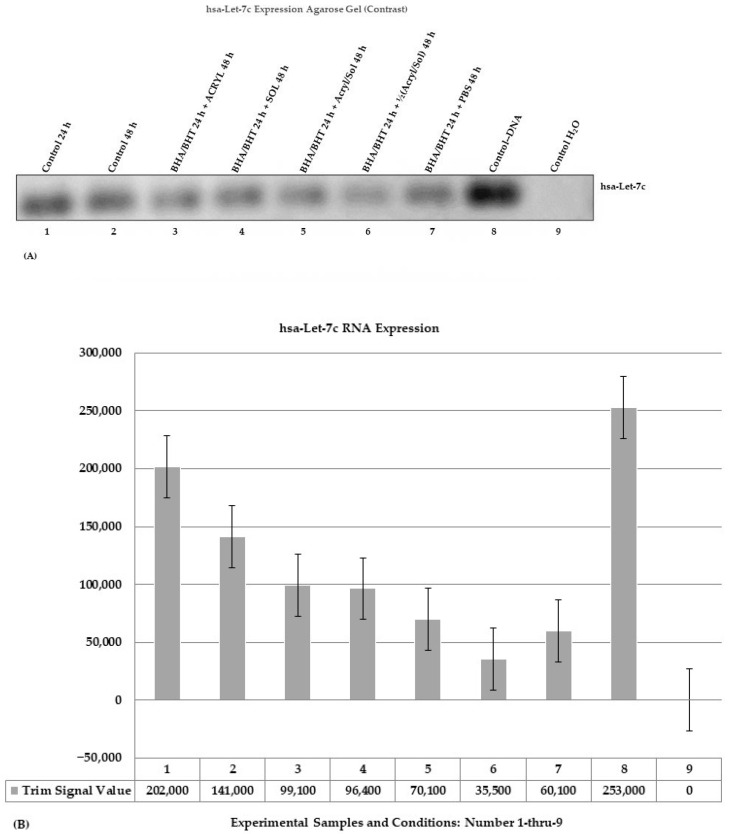
Changes in RNA contents determined by PCR: (**A**) A contrast image of an agarose electrophoresis gel with bands of amplified has-Let-7c stem-loop PCR products analyzed on a 2.5% TAE agarose gel stained with ethidium bromide. The amplified hsa-Let-7c stem-loop bands were organized and ordered in an agarose gel, as was the corresponding bar graph (1–9). The positions of the controls (1, 2, 8–9) and experimental samples (3–7) are numbered. (**B**) Bar graph with standard error bars of amplified hsa-Let-7c stem-loop bands with trim signal values.

**Figure 4 ijms-24-11956-f004:**
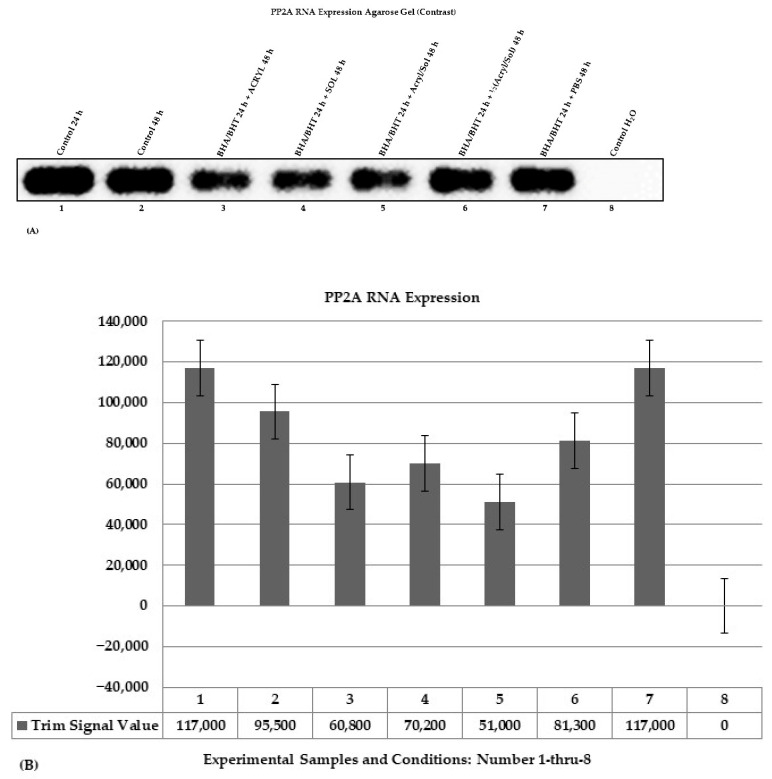
Changes in RNA contents determined by PCR. (**A**) A contrast image with bands of amplified PP2A PCR products arranged (1–8) and analyzed on a 2.5% TAE agarose electrophoresis gel stained with ethidium bromide. (**B**) A bar graph with standard error bars and trim signal values of the amplified PP2A bands. The trim signal values were used to quantify the amount of PP2A expression. The positions of the controls (1, 2, 7, and 8) and experimental samples (3–6) are shown in Figure 4A,B.

**Figure 5 ijms-24-11956-f005:**
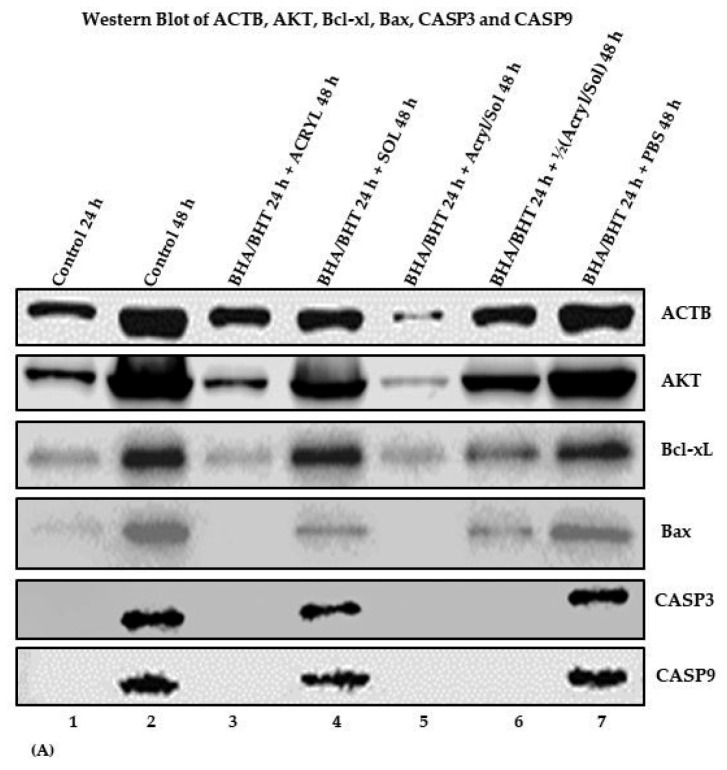
(**A**) Western blots: ACTB, AKT, Bcl-xL, Bax, CASP3, and CASP9 (**B**) A bar graph with error bars and a numerical table of trim signal values for each protein at different conditions.

**Table 1 ijms-24-11956-t001:** Data Summary.

Groups	N	Mean	Std. Dev.	Std. Error
AKT	7	1.1483	0.4514	0.1706
Bcl-xL	7	2.8488	5.4624	2.0646
Bax	7	0.3373	0.314	0.1187
CASP3	7	0.2486	0.3725	0.1408
CASP9	7	0.179	0.2327	0.0879

**Table 2 ijms-24-11956-t002:** One-way ANOVA Summary.

Source	Degrees of Freedom	Sum of Squares	Mean Square	F-Stat	*p*-Value
	DF	SS	MS		
Between Groups	4	35.7458	8.9364	1.4731	0.2351
Within Groups	30	181.9985	6.0666		
Total:	34	217.7442			

## Data Availability

The data obtained for this study are available by contacting the PI’s. The information about the primer sets for the genes is available on the UCSC Genome Browser website.

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
