# Peer review of "Antioxidants Amelioration Is Insufficient to Prevent Acrylamide and Alpha-Solanine Synergistic Toxicity in BEAS-2B Cells"

_ijms, 2023, doi:10.3390/ijms241511956_

Round 1

Reviewer 1 Report

1.     Page 1, Line 21, “α-solanine (0.044 M)” should be amended “α-solanine (44 mM)”.

2.     Page 1, Line 37, “cellular functions [1, 2, 3].” should be amended “cellular functions [1‒3].”.

3.     Page 1, Line 30, “alpha (α)-solanine” should be amended “α-solanine”.

4.     Page 2, Line 54, “a higher temperature [9, 10, 11, 12, 13].” should be amended “a higher temperature [9‒13].”.

5.     Page 6, 7, Line 260, 295-298, “Acylamide (3.5 mM) and ?-Solanine (0.044 260 M) or Acrylamide ½(3.5 mM) and ?-Solanine ½ (0.044 M)” should be amended “acylamide (3.5 mM) and ?-Solanine (44 mM) or acrylamide ½ (3.5 mM) and ?-Solanine ½ (44 mM))”.

6.     Page 10, Line 430, “formation [61, 62, 63, 64]” should be amended “formation [61‒64]”.

7.     Page 10, Line 409, “[29, 54, 55, 56, 57]” should be amended “[29, 54‒57]”.

8.     The format and arrangement in the “References” should conform to “International Journal of Molecular Sciences”.

International Journal of Molecular Sciences, Manuscript Number: ijms-2474351-peer-review-v1

The manuscript I have reviewed is about the “Antioxidants Amelioration Is Insufficient To Prevent Acrylamide and Alpha-solanine Synergistic Toxicity in BEAS-2B Cells”.  We were marked the error using the red words on the manuscript and provided to you.

Comments:

1.     Page 1, Line 21, “α-solanine (0.044 M)” should be amended “α-solanine (44 mM)”.

2.     Page 1, Line 37, “cellular functions [1, 2, 3].” should be amended “cellular functions [1‒3].”.

3.     Page 1, Line 30, “alpha (α)-solanine” should be amended “α-solanine”.

4.     Page 2, Line 54, “a higher temperature [9, 10, 11, 12, 13].” should be amended “a higher temperature [9‒13].”.

5.     Page 6, 7, Line 260, 295-298, “Acylamide (3.5 mM) and ?-Solanine (0.044 260 M) or Acrylamide ½(3.5 mM) and ?-Solanine ½ (0.044 M)” should be amended “acylamide (3.5 mM) and ?-Solanine (44 mM) or acrylamide ½ (3.5 mM) and ?-Solanine ½ (44 mM))”.

6.     Page 10, Line 430, “formation [61, 62, 63, 64]” should be amended “formation [61‒64]”.

7.     Page 10, Line 409, “[29, 54, 55, 56, 57]” should be amended “[29, 54‒57]”.

8.     The format and arrangement in the “References” should conform to “International Journal of Molecular Sciences”.

Author Response

All edits made according to suggestion. Thank you for your time.

Reviewer 2 Report

The work is very interesting however I have two important comments:
1. Why in the Western blot method the control protein is not homogeneous in the samples? and the differences are significantly large
2. I think that on the graphs should be ststistics: deviation and significance p<0.05
Please clarify these important points.

Author Response

Point 1:Why in the Western blot method the control protein is not homogeneous in the samples? and the differences are significantly large

Response 1: The toxic chemicals we use have an effect on control protein as well as all other proteins due to cell death and adduct formation. Give this no matter what the control protein is there will be a variation observed in the control protein similar to sodium arsinite effect.

Point 2: I think that on the graphs should be statistics: deviation and significance p<0.05

Response 2: This has been added

Thank you for your time and we look forward to your aid during this process. 

Reviewer 3 Report

Overall, interesting topic. Poorly written, very poor figures/tables and lacks a well presentation and many things are not clear also there is a lot of repetition. The main point what is the novelty of this work? Please state this clearly

Line 496: two million cells. How you obtained this number based on what?

Figure 1: Really unclear it’s not in the level of publication!? Since it’s very difficult to judge he scientific content when it is presented in this way! May I ask how you could distinguish between green, red and orange!?

Table 1 and 2. Unclear. Improve please

Figure 3, 4 and 5. Unreadable

All gel images are unclear!

It is difficult to scientifically judge while the manuscript in this form.

English check required

Author Response

Point 1: The main point what is the novelty of this work? Please state this clearly

Response 1: Potatos, a vastly consumed food has harmful byproducts as well as antioxidants thought to medigate there effects. In this study we attempt to understand the interplay of both the byproducts and the antioxidant on cells. To our knowledge, this study provides the first evidence that shows synergistic effects of two of the byproducts (acrylamide and α-solanine) in the presence of antioxidants in BEAS-2B cells. This study is novel in that it exposes the inability of the antioxidants to completely hinder the toxicity when the byproducts are in combination. This is cause for concern and further studies are needed to better understand these efforts being both of these bypoducts are found in potatos and they are vastly consumed in high quantities.

Point 2: Line 496: two million cells. How you obtained this number based on what?

Response 2: Cells were counted via BioRad TC20 Automated Cell Counter and seeded at two million cells per culture dish plate. We conducted an experiment prior to this one to establish a laboratory standard for 80% confluency being an estimated 2million cells. We included an additional conting control that was not mentioned in the method section for verification because we did not want the trypan blue to interfer with our experiments. The 60x15 plates used can hold 2.5 million cells.

Point 3: Figure 1: Really unclear it’s not in the level of publication!? Since it’s very difficult to judge he scientific content when it is presented in this way! May I ask how you could distinguish between green, red and orange!?

Response 3: Images were taken at 10x magnification and cell morphology assesed while viewing. Crenated cells display an elongated cell body as depicted very clearly in 1E (green arrow). Shrunken cells are commonly darker pigmented and irregular in shape as seen in 1C (red arrow). While apoptotic cells have two forms fragmentation as seen in 1F (orange arrow) and apoptotic bodies in 1E (orange arrow). Image arrows were then placed for publication accordingly.

Point 4: Table 1 and 2. Unclear. Improve please; Figure 3, 4 and 5. Unreadable; All gel images are unclear!

Response 4: Revisions have been made to all figures mentioned.

Thank you for your time and I look forward to working with you in the near future

Round 2

Reviewer 2 Report

I believe that the quality of the presented western blot results is not adequate, and I stand by my opinion on this issue should be improved. In addition, statistics have not been entered on every chart.

Author Response

1. First, we thank the reviewer for commenting on the "Western blot results inadequate." The reviewer's comment was well received, and we have fixed the Western blot issues. Also, we thank the reviewer for the earlier statement or observation that the work was interesting.

2. Second, concerning the statistics, we mistakenly thought the reviewer only requested error bars for Figure 5B, the bar graph for the Western blot, since it was unclear to us in the reviewer's first comment. We have placed error bars on all the bar graphs, Figures 3B, 4B, and 5B.

3. Finally, we thank the reviewer for the comments and improvement of the manuscript.

Reviewer 3 Report

Thanks for the revised version, although my main concern still… I don’t get what would be next clearly? Moreover, what I see in this manuscript is not novel and that’s why I mentioned please clarify this point... The Very few techniques used in this study not enough to make this interesting conclusion.

What is the difference between One-way ANOVA, two or T-TEST? please justify your selection since if we move one to another you might get different results

Figures still in very bad condition example (Page 5 *Very hard to see the morphological changes since what you did is just zooming in which made the situation worse*, page 9, 10 and 12 please find different way to present the results I don’t think it is in the level of publishing)  

I think It would be great to use different viability assays to validate more plus to look for the metabolic finger prints in this way you could validation in correlates your results.

Good luck.

Author Response

1. First, we thank the reviewer for commenting on the "Test statistic." The reviewer's comment was well-received. Three tests of statistical significance will produce different results since they are based on probability that tests the relationship between two variables. Which of the three tests of statistical significance a person or an investigator choose depend on the statistical questions and data collected by a person or an investigator, meaning which statistic is best fit to analyze the data. Generally, ANOVA is used to compare the means among three or more groups, and it is a one-side (one-tail) test. The t-test or student's t-test compares the means of two groups and is a two-sides (two-tail) test. In addition, the t-test is unsuitable for analyzing our data for the "Null hypothesis." One-way ANOVA has one independent variable, while two-way ANOVA has two independent variables and assesses the interrelationship of two independent variables on a dependent variable.

2. Second, We have modified the presentation of the results on pages 9, 10, and 12, taken the reviewer's suggestions seriously, and made adjustments as fit.

3. Third, it is well taken for the different viability assays to validate more and look for the metabolic fingerprints suggested by the reviewer. Currently, we still need to be set up to perform metabolic fingerprints. We will consider it as our laboratory research interests and goals progress.

4. Finally, we thank the reviewer for the comments and improvement of the manuscript.

Round 3

Reviewer 2 Report

The Auror answers questions about the issues and that the work can be published.